# Predicting Quality of Life in Women with Breast Cancer Who Engage in Physical Exercise: The Role of Psychological Variables

**DOI:** 10.3390/healthcare11142088

**Published:** 2023-07-21

**Authors:** Santiago Fresno-Alba, Marta Leyton-Román, Sara Mesquita da Silva, Ruth Jiménez-Castuera

**Affiliations:** 1Gynecology and Obstetrics Service University Hospital San Pedro de Alcántara of Cáceres, Faculty of Sport Sciences, University of Extremadura, 10003 Cáceres, Spain; santiago.fresno@salud-juntaex.es; 2Faculty of Teacher Training, University of Extremadura, 10003 Cáceres, Spain; 3Faculty of Sport, University of Porto, 4099-002 Porto, Portugal; saramesquita@fade.up.pt; 4Faculty of Sport Sciences, University of Extremadura, 10003 Cáceres, Spain; ruthji@unex.es

**Keywords:** breast cancer, motivation, physical activity

## Abstract

In this study, we aimed to conduct a descriptive analysis of the primary physiological and psychological factors influencing the quality of life in women with breast cancer who engage in physical exercise. The study examined the key psychological variables predicting patients’ quality of life, perceived support from family and friends, and the perception of physical condition. The sample consisted of 46 women from Cáceres (Spain) aged between 30 and 75 years undergoing breast cancer treatment. The Functional Evaluation Scale in Cancer Therapy (FACT-B+4) was used to measure quality of life; the Basic Psychological Needs Satisfaction Scale was used to measure autonomy, competence, and social relationships; the Behavior Regulation Questionnaire in Exercise (BREQ-3) was used to measure the types of self-determined motivation for sports participation; the General Evaluation of Self-Esteem Scale was used to measure self-esteem; the International Fitness Scale (IFIS) was used to measure perceived physical condition; and the Perceived Autonomy Support Scale for Exercise Settings (PASSES) was used to measure the perceived autonomy support from family and friends. A multiple regression analysis revealed that perceived physical condition and self-esteem were significant positive predictors of a variance in quality of life, while intrinsic motivation did not significantly predict it. The findings underscore the importance of promoting autonomous motivation in patients to enhance their physical and psychological well-being through physical activity.

## 1. Introduction

Breast cancer (BC) is the most prevalent form of cancer in women [1] and is recognized as one of the leading causes of mortality among women aged 30 to 54 years [2]. In Spain, approximately 0.12% of females between 15 and 65 years of age are affected by breast cancer, while 0.24% of women over 65 are affected, with a 7.5% increase observed between 2012 and 2019 [3]. Risk factors for breast cancer can be categorized into three groups: modifiable, non-modifiable, and protective factors. The primary non-modifiable risk factors include age (with an increased incidence after 35 years and stabilization at 55 years), family medical history [4], and physiological hormonal status [5].

The modifiable risk factors for breast cancer include the following: childbirth (which reduces the risk by 10%) [5], breastfeeding (which reduces the risk by 2% for every 5 months of breastfeeding) [6], and alcohol consumption (which increases the risk by up to 30%) [7]. Furthermore, although the evidence is limited, physical activity (PA) [8], nutrition [9], and screening [10] have been associated with protective effects against breast cancer. When the disease is diagnosed, the primary treatments for breast cancer involve systemic approaches such as hormonal therapy, chemotherapy, and biological agents, which may be used in combination with local treatments like surgery and radiotherapy [11].

Cancer is associated with significant complications such as muscle atrophy, weakness, and weight gain, leading to a poor quality of life [12,13]. These adverse effects are also a result of physical inactivity, which reduces physical function, aerobic capacity, and quality of life [12]. In this regard, there is a growing body of scientific evidence on the physical and psychological benefits of exercise during and after cancer treatment [14]. Several studies [15,16,17,18] have demonstrated that exercise can have positive effects on patients during treatment, including improvements in aerobic fitness, upper and lower body strength, body weight, fat percentage, quality of life, mental state, mood, anxiety, self-esteem, treatment efficacy, immunity, and bone health. Furthermore, exercise has been shown to have benefits for patients after treatment, including the aforementioned improvements as well as with respect to body mass index, level of physical exercise, fatigue, general symptoms and side effects, relapse rates, survival rates, and life expectancy [15,16,17,18]. Consequently, engaging in regular physical activity is not only associated with an increase in post-diagnosis quality of life but also a reduction in treatment side effects [19,20]. In summary, there is evidence to suggest that regular physical activity following recommended guidelines reduces the risk of cancer recurrence and mortality both before diagnosis and during and after treatment [21].

The motivation of patients towards physical activity is a central aspect of the present study. We believe that applying theoretical frameworks can help us to understand the cognitive and motivational processes related to physical activity and develop interventions to promote it. In this study, we adopted the self-determination theory (SDT), which is widely regarded as one of the most valid motivational theories currently available [22,23]. According to SDT, motivation for physical activity is determined by the satisfaction of three fundamental psychological needs (BPNs) during practice: competence, autonomy, and relatedness. Competence refers to the perceived effectiveness and ability to accomplish planned activities. Autonomy involves the freedom to make choices and decisions throughout the process. Relatedness encompasses efforts to connect with and care about others, as well as to experience authentic relationships and satisfaction within a social context. When these BPNs are satisfied during physical activity, individuals are more likely to exhibit self-determined forms of motivation such as intrinsic motivation (engaging in the activity for pleasure and enjoyment), integrated regulation (incorporating the activity into one’s lifestyle), and identified regulation (recognizing the activity’s importance and its benefits, such as for health). Conversely, when these BPNs are frustrated, non-self-determined forms of motivation can arise, including introjected regulation (performing the activity due to feelings of guilt), external regulation (engaging solely to obtain external rewards or due to pressure from others), and amotivation (lack of intention to engage in the activity) [22].

Vallerand [24,25] expanded on the assumptions of the self-determination theory (SDT) and developed the hierarchical motivation model. This model represents a significant advance in understanding motivational processes. According to this model, the satisfaction of the BPNs leads to varying levels of self-determined motivation, which in turn influence affective, cognitive, and behavioral outcomes. The nature of these outcomes, whether positive or negative, depends on the degree of self-determination achieved by the individual. Specifically, in the context of cancer patients, this is if they attain the highest levels of self-determined motivation. This condition is associated with positive consequences such as improved adherence to physical activity, enhanced performance, and various indicators of well-being [26].

A limited number of studies have applied the theoretical framework of SDT to investigate motivation towards physical exercise, specifically in breast cancer patients [27,28,29,30]. Milne et al. [27] found that women survivors of breast cancer who adhered to physical activity guidelines exhibited significantly higher levels of intrinsic motivation, identified regulation, and satisfaction of the basic psychological need for competence. Hawkins et al. [28] implemented a randomized intervention program for six months, utilizing SDT mediators through internet and telephone communications, which resulted in improved quality of life for breast cancer patients. Hull et al. [29] assessed the effectiveness of a comprehensive support system incorporating the three basic psychological needs along with an interactive web-based platform to enhance quality of life in breast cancer patients. Jimenez et al. [30] conducted descriptive and linear regression analyses of key physiological and psychological variables influencing the quality of life in breast cancer patients participating in a three-month physical intervention program.

The objectives of this study were to determine the variables that predict the quality of life of women with breast cancer and to describe the relationships between the psychological variables analyzed.

## 2. Materials and Methods

This study received ethical approval from the Commission of Bioethics and Biosecurity at the University of Extremadura, Spain, and adhered to the Helsinki Declaration guidelines. Participants were treated in accordance with the ethical guidelines of the American Psychological Association (APA), including obtaining participant assent, parent/guardian consent, and ensuring confidentiality and anonymity. Informed written consent was obtained from all participants.

### 2.1. Sample

The study sample consisted of 46 women aged between 30 and 75 years (*M* = 52.20; *SD* = 11.37) who had recently undergone breast cancer surgery at the Hospital San Pedro de Alcántara in Cáceres, Spain. All participants were undergoing chemotherapy, radiotherapy, and/or hormone treatment at the time of the study. A total of 77% of participants independently engaged in moderate physical activity, with an average of 3 h of exercise per week.

Participants were intentionally selected for inclusion in the study following the methodology described by Montero and León [31]. Exclusion criteria for participation in the study included: (1) known cardiac abnormalities such as unstable angina or recent myocardial infarction; (2) significant physical disabilities affecting physical function, including severe arthritis; (3) a current diagnosis of a serious psychiatric illness (participants with minor psychiatric diagnoses were eligible if well enough to participate); and (4) enrollment in a trial or behavioral health program.

### 2.2. Variables and Measures

The variables and measurement instruments used in the study were as follows.

***Quality of life (dependent variable).*** The specific measure for breast cancer of the Functional Evaluation Scale in Cancer Therapy (FACT-B+4) was used, which is based on the original FACT scale developed by Cella et al. [32]. The Spanish version of this instrument was validated by Martínez et al. [33]. The FACT-B+4 consists of 42 items that assess different factors related to quality of life in breast cancer patients, including physical well-being (7 items, e.g., “I experience pain”), social/family well-being (7 items, e.g., “My family has accepted my illness”), emotional well-being (6 items, e.g., “I feel sad”), and functional well-being (7 items, e.g., “I am able to enjoy life”). The scale also has items that measure breast-cancer-specific issues (10 items, e.g., “I am bothered by my hair loss”), and lymphedema-specific issues (5 items, e.g., “Moving my arm on that side causes me pain”). Participants responded to all items using a Likert-type scale ranging from 1 (not at all) to 5 (very much) to indicate their level of agreement or experienced impact. For the present study, the FACT-G total score was used as the dependent variable. FACT-G was calculated by forming the mean average score for the four well-being scales and then summing them together. The higher the score, the higher the quality of life (possible score range = 0–108).

***Basic Psychological Need Satisfaction.*** The Basic Psychological Need Satisfaction Scale developed by Wilson et al. [34] and validated in Spanish by Moreno-Murcia et al. [35] was used to assess participants’ satisfaction of basic psychological needs. The scale consists of 18 items organized into three factors, each introduced with the phrase “In my training…”. These are autonomy, comprising 6 items (e.g., “I think I can choose the exercises in which I participate”); competence, comprising 6 items (e.g., “I feel capable of completing the most challenging exercises”); and relatedness, comprising 6 items (e.g., “I think I get along well with my classmates when we do exercises together”). Participants rated each item on a Likert-type scale ranging from 1 (false) to 5 (true).

***Types of self-determined motivation.*** To assess motivation towards physical activity, the Behavior Regulation Questionnaire in Exercise (BREQ-3) developed by Wilson et al. [36] and validated for the Spanish context by González-Cutre et al. [37] was utilized. This questionnaire comprises 23 items designed to measure various types of self-determined motivation. The items are categorized into six factors and are introduced with the phrase “I do physical exercise…”. These are intrinsic regulation, consisting of 4 items (e.g., “Because I believe that exercise is fun”); integrated regulation, consisting of 4 items (e.g., “Because it aligns with my way of life”); identified regulation, consisting of 3 items (e.g., “Because I recognize the benefits of physical exercise”); introjected regulation, consisting of 4 items (e.g., “Because I feel guilty when I don’t do it”); external regulation, consisting of 4 items (e.g., “Because others tell me I should do it”); and demotivation, consisting of 4 items (e.g., “I don’t see the point of exercising”). Participants provided responses on a Likert-type scale ranging from 1 (not at all true) to 5 (totally true).

***Self-esteem.*** The scale used to measure self-esteem was the General Evaluation of Self-Esteem Scale developed by Rosenberg [38] and validated in Spanish by Vázquez-Morejón et al. [39]. This unidimensional instrument comprises 10 items, with 5 items positively formulated and 5 items negatively formulated. The scale assesses a single factor, self-esteem, and measures participants’ perceptions of their self-worth (e.g., “I feel that I have a number of good qualities”). Responses to all items were provided on a Likert-type scale ranging from 1 (totally disagree) to 5 (totally agree).

***Perceived physical condition.*** The perceived physical condition of participants was assessed using the International Fitness Scale (IFIS) developed by Ortega et al. [40]. This scale has 5 items that measure the individual’s perception of their overall physical condition (e.g., “My general physical condition is…”). Participants rated each item on a Likert scale ranging from 1 (very bad) to 5 (very good). The IFIS [40] has demonstrated a strong correlation with objective measures of physical condition in previous research.

***Autonomy support.*** The perceived autonomy support for physical activity among patients regarding their family and friends was assessed using the Perceived Autonomy Support Scale for Exercise Settings (PASSES) developed by Hagger et al. [41]. The scale consists of 12 items that specifically address the support received from family and friends in relation to physical exercise (e.g., “My family or friends understand why I decide to exercise during treatment”). Participants rated each item on a Likert scale ranging from 1 (totally disagree) to 7 (totally agree).

### 2.3. Procedure

The principal investigator of the study established contact with the Department of Oncology and Gynecology at Hospital San Pedro de Alcántara in Cáceres, Spain, seeking permission to involve breast cancer patients in the research. The hospital was duly informed about the study and consent was obtained to include eligible patients. The specialist physician in charge of the patients diagnosed with breast cancer informed them about the study if they met the inclusion criteria. Patients who expressed interest in participating provided informed consent and completed the aforementioned questionnaires. The principal investigator supervised the process. The time required to complete the questionnaires was approximately 20 min.

### 2.4. Data Analysis

Descriptive statistics, including means and standard deviations as well as indicators of skewness and kurtosis, were computed for the study variables (Table 1). This analysis allowed for a comprehensive understanding of the central tendencies, variability, and the shape of the distributions of the variables. To evaluate the reliability of the study measures, McDonald’s omega coefficients with 95% confidence intervals were calculated for each [42] using JASP (version 0.16.4.0; JASP Team, 2023) (Table 1).

As a preliminary step for the regression analysis, Pearson’s correlations were computed among all study variables, providing an assessment of their inter-relationships and assisting in identifying potential predictors for the subsequent regression model (Table 2). Guided by the results of this correlational analysis, a multiple linear regression was performed, including intrinsic motivation, self-esteem, and perceptions of physical fitness as predictors of the FACT-G total. Basic needs and identified regulation were not considered in the final model due a pattern of non-significant correlations. Moreover, integrated regulation, introjected regulation, external regulation, and amotivation were excluded to avoid multicollinearity.

The Durbin–Watson value for this model was 1.85, supporting independence of model residuals. Variance inflation factor (VIF) values were all substantially lower than 10, supporting an absence of multicollinearity among predictors (Table 3). Furthermore, an inspection of a residual plot was supportive of normality. Finally, the F-test for heteroskedasticity was non-significant (*p* = 0.065), suggesting homogeneity of variance. Thus, the analysis was performed without bootstrapping or robust SE estimators. Given the small sample size, a more liberal alpha was set (α = 0.10) to assess the statistical significance [43]. However, because this value was not set a priori, we considered such a finding of ‘marginal significance’ relative to the more traditional alpha of 0.05. This regression analysis was performed using SPSS 27.0.

## 3. Results

### 3.1. Descriptive Analysis

The means, standard deviations, and estimates of skew and kurtosis for all study variables are shown in Table 1. Notable from this table was that all values of skew and kurtosis were below traditional thresholds for acceptability (values < 2.0), indicating no major deviations from normality.

### 3.2. Correlational Analysis

The pattern of correlations identified from the correlational analysis was generally consistent with theoretical expectations (Table 2). Notably, intrinsic motivation and integrated motivation in exercise were positively associated with the FACT-G total score, while introjected regulation, external regulation, and amotivation in exercise all expressed negative correlations. Self-esteem also showed a positive correlation with FACT-G, albeit not significant and at a lower magnitude than intrinsic and integrated motivation. Basic need satisfaction did not appear to have a meaningful association with the FACT-G total score. Finally, participants’ perceptions of their physical condition were strongly correlated with FACT-G.

### 3.3. Linear Regression

The model accounted for a significant proportion (43.7%) of variance in the FACT-G total. The R^2^ adjusted = 0.40, *F*(3,42) = 10.85, and *p* < 0.001. Standardized beta coefficients indicated that perceived physical condition was the strongest predictor (β = 0.36; *p* = 0.016), followed by self-esteem (β = 0.30; *p* = 0.014) and intrinsic motivation (β = 0.26; *p* = 0.070).

## 4. Discussion

The current study aimed to ascertain the psychological factors associated with the quality of life among women diagnosed with breast cancer. The correlation results indicated that intrinsic motivation and integrated motivation in exercise were positively associated with the FACT-G total score, while introjected regulation, external regulation, and amotivation in exercise all expressed negative correlations. Self-esteem also showed a positive correlation with FACT-G, albeit not significant and at a lower magnitude than intrinsic and integrated motivation. Basic need satisfaction did not appear to have a meaningful association with the FACT-G total score. Finally, participants’ perceptions of their physical condition were strongly correlated with FACT-G. Regarding regression, the results indicated that perceived physical condition, self-esteem, and intrinsic motivation emerged as significant predictors of quality of life.

### 4.1. Relevant Psychological Variables

Individuals diagnosed with cancer often experience significant changes in their personal lives, including a lack of intrinsic motivation due to environmental deprivation and potential disruptions to their family and social contexts [44]. In addressing this issue, Bruno and Theran [45] proposed implementing interventions aimed at enhancing the motivation of individuals with breast cancer. Among these strategies, engaging in physical activity has been widely demonstrated as a beneficial approach [46,47].

Intrinsic motivation plays a crucial role in promoting adherence to physical activity among breast cancer patients. Several authors [28,46,47,48,49] have highlighted that improved quality of life in breast cancer patients is associated with the physical and psychological benefits derived from regular physical activity. This can be achieved through various approaches such as supervised group programs [50] as well as telephone and online support [51,52]. However, in our current study, intrinsic motivation did not emerge as a significant predictor of quality of life in breast cancer patients at the traditional alpha level of *p* < 0.05 (although it could be considered to be marginally significant at the less conservative level of *p* < 0.10). This finding could be attributed to the fact that the participants had not yet initiated a physical exercise program despite their potential interest. It is possible that conducting a longitudinal analysis following an intervention program in physical exercise would yield different results.

Along the same lines, Fu et al. [53] conducted a study utilizing a structural equation analysis and found that, among individuals with breast cancer undergoing chemotherapy treatment, a lack of motivation (demotivation) to undertake physical activity and towards satisfying certain basic psychological needs (BPNs) such as feelings of competence and autonomy were commonly observed.

The present study aligned with these findings, revealing a positive and significant relationship between competence and integrated regulation, which represents one of the most self-determined forms of motivation. Consequently, it is imperative to prioritize the implementation of strategies aimed at enhancing competence, thereby fostering intrinsic motivation among women with breast cancer in order to facilitate long-term improvements in their quality of life.

The study’s findings suggested that women with breast cancer had limited confidence and understanding of the benefits of physical activity programs. Effective communication by professionals and the use of objective tests have been shown to improve this situation and enhance intrinsic motivation towards such programs [54]. Various physical activity programs have been developed for women with breast cancer [55,56,57] that employ motivational strategies to promote adherence and enhance quality of life. Therefore, further research in this field should focus on advancing interventions for this specific population [54].

Physical activity programs have been found to positively impact various aspects of quality of life in breast cancer patients, including their perception of themselves. This includes improvements in physical condition, state of mind, and self-esteem [16].

Self-esteem plays a significant role in breast cancer patients as it often diminishes as a result of the condition [58]. To address this, studies like Musanti [59] and Patsou et al. [60] incorporated exercises aimed at improving physical and overall self-esteem, leading to increased adherence to the practice and improved quality of life. In the present study, self-esteem was identified as a significant predictor of quality of life in women with breast cancer. 

Numerous studies have demonstrated that enhancing the physical condition of women with breast cancer through physical exercise positively impacts their quality of life [61]. These findings align with the current study, which showed a positive and significant correlation between perceived physical condition, quality of life, and self-determined motivation. Moreover, qualitative studies [62] have revealed that fatigue from treatments and psychological distress due to changes in body image perception negatively impact the quality of life of women with breast cancer, further supporting the importance of these results. Consequently, it is recommended that physical activity is initiated soon after a breast cancer diagnosis to enhance self-esteem and improve physical condition perception [63], thereby optimizing quality of life [60].

### 4.2. Limitations and Future Efforts

The study’s limitations include the subjective nature of the measurements and its descriptive design. Although descriptive studies are crucial to establish baseline information and understand variable behaviors, future research should focus on intervention programs that incorporate guided physical activity sessions and strategies to enhance self-determined motivation, self-esteem, and body image perception. Objective measures, including biological and physical condition variables, should be incorporated alongside psychological variables to comprehensively evaluate patients from various perspectives. Furthermore, it would be valuable to replicate this study after implementing a physical exercise program to determine which variables are most influential in patients’ quality of life, both before and after the initial diagnosis and therapy (e.g., radiotherapy, chemotherapy, and hormone therapy). Finally, we acknowledge that the sample size for the study was relatively small. Estimated *p*-values are dependent on the sample size, meaning that when sample sizes are small, the likelihood of committing a Type II error is heightened. Although we attempted to counter this using a more liberal alpha threshold to determine significance, it is possible that the analytical approach lacked statistical power to identify some true population effects. Thus, future studies with larger samples are required to replicate and confirm these findings.

This research underscores the importance of enhancing self-esteem and physical condition perception in the early stages following a breast cancer diagnosis. It is crucial to encourage patients to engage in physical exercise programs that foster self-determined motivation, aiming to improve their quality of life. To achieve this, strategies should be employed to address autonomy, competence, and social relationships within physical activity sessions. Professionals should adopt an understanding-based approach, actively responding to the interests, opinions, and emotions of women. Encouraging decision-making, involving patients, and assigning them responsibilities fosters a sense of autonomy. Group work and cohesion should be promoted to facilitate social connections. Additionally, providing opportunities for patients to lead parts of the session or activity and establishing different levels of task difficulty will encourage effort and personal progress, thereby improving both their competence and perception of physical condition and self-esteem.

## 5. Conclusions

In the regression, perceived physical condition emerged as the most significant predictor of quality of life, followed by self-esteem. Although not reaching a statistical significance, intrinsic motivation towards physical activity practice also played a role. Therefore, physical exercise programs designed for breast cancer patients should prioritize the promotion of these variables to enhance their quality of life.

## Figures and Tables

**Table 1 healthcare-11-02088-t001:** Descriptive statistics.

					Omega
	*M* ^1^	*SD* ^2^	Skewness	Kurtosis	Estimate, 95% CI
FACT-G Total	75.74	16.82	−0.08	−0.81	0.90 [0.85, 0.93]
Intrinsic Motivation	3.83	0.96	−0.60	−0.54	0.80 [0.68, 0.89]
Integrated Regulation	3.68	1.03	−0.60	−0.20	0.87 [0.80, 0.93]
Identified Regulation	4.22	1.03	−1.59	1.94	0.90 [0.83, 0.94]
Introjected Regulation	2.30	0.95	0.15	−0.94	0.70 [0.57, 0.82]
External Regulation	1.93	1.03	0.69	−0.70	0.78 [0.65, 0.90]
Amotivation	1.80	0.96	1.00	−0.20	0.81 [0.68, 0.90]
Competence	3.64	0.97	−0.49	−0.07	0.90 [0.82, 0.95]
Autonomy	4.04	0.92	−1.07	1.34	0.89 [0.78, 0.94]
Relatedness	3.33	0.97	−0.53	−0.23	0.85 [0.77, 0.91]
Self-Esteem	4.24	6.91	−0.53	−0.58	0.81 [0.70, 0.88]
Perceived Physical Condition	2.92	0.86	−0.08	0.12	0.89 [0.82, 0.94]

^1^ Mean; ^2^ standard deviation.

**Table 2 healthcare-11-02088-t002:** Correlational analysis.

	1	2	3	4	5	6	7	8	9	10	11	12
1. FACT-G Total	1											
2. Intrinsic Motivation	0.48 ***	1										
3. Integrated Regulation	0.43 **	0.79 ***	1									
4. Identified Regulation	0.29	0.48 ***	0.52 ***	1								
5. Introjected Regulation	−0.42 **	−0.10	−0.05	−0.14	1							
6. External Motivation	−0.49 ***	−0.31 *	−0.16	−0.23	0.61 ***	1						
7. Amotivation	−0.44 **	−0.38 *	−0.40 **	−0.58 ***	0.44 **	0.59 ***	1					
8. Competence	0.21	0.36 *	0.57 ***	0.30 *	−0.14	−0.04	−0.19	1				
9. Autonomy	0.18	0.21	0.41 **	0.28	−0.06	0.03	−0.23	0.70 ***	1			
10. Relatedness	0.13	0.25	0.27	−0.01	0.27	0.21	−0.01	0.40 **	0.21	1		
11. Self-Esteem	0.38 *	0.07	0.18	0.41 ***	−0.48 ***	−0.31 *	−0.45 **	0.11	0.16	−0.44 **	1	
12. Perceived Physical Condition	0.55 ***	0.56 ***	0.53 ***	0.33 *	−0.11	−0.24	−0.16	0.39 **	0.14	0.29	0.17	1

** p* < 0.05; ** *p* < 0.01; *** *p* < 0.001.

**Table 3 healthcare-11-02088-t003:** Output from multiple linear regression.

					90% CI for *b*	
	*b*	*SE*	*t*	*p*	LB	UB
(Intercept)	8.28	13.53	0.61	0.544	14.48	31.03
Intrinsic Motivation	4.58	2.46	1.86	0.070	0.44	8.72
Self-Esteem	0.74	0.29	2.57	0.014	0.25	1.22
Perceived Physical Condition	6.96	2.78	2.50	0.016	2.27	11.64

Dependent variable: FACT-G total.

## Data Availability

The data presented in this study are available in the Appendix A.

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
