# Peer review of "Predicting Quality of Life in Women with Breast Cancer Who Engage in Physical Exercise: The Role of Psychological Variables"

_healthcare, 2023, doi:10.3390/healthcare11142088_

Round 1

Reviewer 1 Report

This is a descriptive analysis that aims to determine the primary physiological and psychological factors influencing the QoL in women with breast cancer who engage in physical exercise.

Abstract:

Abstract is well written. May be helpful to specify what geographical region the study sample was taken from. Also need to improve methods section (ie. what measures/outcomes were administered to patients), and it may be helpful to put it after introducing participant demographic characteristics.

Introduction:

Clear background information and rationale given for undertaking the study. Good work explaining all the modifiable and non-modifiable risk factors for breast cancer. Section describing the specific theoretical framework can be better used in the Methods section.

Methods:

Methods section is written clearly, especially when explaining the questionnaires/scales administered to patients.

Results:

May be useful to add a section on patient characteristics at the beginning of the results section. May also be helpful to summarize significant results in a coherent manner in words, instead of just through the tables.

Discussion/Conclusion:

First section of the discussion section should summarize the results of the study in more detail. May also be helpful to extend the results into a local context (ie. Spain) and global context and future directions for research. This section of the manuscript is also well written.

Final Notes: I recommend this paper be submitted for publication, with certain revisions and

additions. The paper presents information on such an important and interesting topic very well.

Minor revision might be needed. 

Author Response

First of all, we would like to thank all the comments and suggestions of both the editor and the reviewer, which helps us to improve and give a higher quality to the work done and presented. Then each comment is answered point by point. We hope to answer cleary to the comments and suggestions of the editor and the reviewer. If it were necessary to clarify any concept or point, let us know.

This is a descriptive analysis that aims to determine the primary physiological and psychological factors influencing the QoL in women with breast cancer who engage in physical exercise.

Abstract:

Abstract is well written. May be helpful to specify what geographical region the study sample was taken from. Also need to improve methods section (ie. what measures/outcomes were administered to patients), and it may be helpful to put it after introducing participant demographic characteristics.

Response: The geographic region to which the women belong has been added (line 19), and information on the measurement instruments and variables (line 20-27).

Introduction:

Clear background information and rationale given for undertaking the study. Good work explaining all the modifiable and non-modifiable risk factors for breast cancer. Section describing the specific theoretical framework can be better used in the Methods section.

Response: Thank you for your input.

Methods:

Methods section is written clearly, especially when explaining the questionnaires/scales administered to patients.

Response: Thank you for your input.

Results:

May be useful to add a section on patient characteristics at the beginning of the results section. May also be helpful to summarize significant results in a coherent manner in words, instead of just through the tables.

Response: We have put the characteristics of the patients in the method section, exactly in point 2.1. Sample (Line 123).

The most relevant results of the correlations are described in Line 244-252, and of the regression in Line 259-262.

Discussion/Conclusion:

First section of the discussion section should summarize the results of the study in more detail. May also be helpful to extend the results into a local context (ie. Spain) and global context and future directions for research. This section of the manuscript is also well written.

Response: We have added at the beginning of the discussion the main findings of the study (Line 265-274).

Final Notes: I recommend this paper be submitted for publication, with certain revisions and additions. The paper presents information on such an important and interesting topic very well.

Response: Thank you so much for your input.

Reviewer 2 Report

See enclosed file.

Only minor grammatical changes required. See enclosed file.

Author Response

First of all, we would like to thank all the comments and suggestions of both the editor and the reviewer, which helps us to improve and give a higher quality to the work done and presented. Then each comment is answered point by point. We hope to answer cleary to the comments and suggestions of the editor and the reviewer. If it were necessary to clarify any concept or point, let us know.

General Comments to Authors:

This study examines the key psychological variables affecting quality of life in women undergoing breast cancer treatment. The statistical techniques used are appropriate, though the study has a modest sample size, slightly too small for the range of variables the authors are exploring. The result is that the authors have derived useful conclusions which have to depend somewhat more on theory than on their empirical findings.

If an alpha value of .10 in the regression cut-off were the only issue, the paper might be able to stand as is. However, the authors do not state whether this alpha critical value was chosen in advance of the analyses; the model R2 adjusted is low (.38); the number of relevant variables has been reduced from 12 to 3 (2 if a more traditional .05 alpha value were used); and the Discussion mentions integrated regulation though it is not a statistically significant regression variable.

The authors can adopt 2 approaches at this point: (1) increase the study N so that a .05 alpha critical value can be used; or (2) reduce their full article to a Brief Report of 2,500 words or less while including the study N as a substantial limitation in the Discussion and stating future work will strive to increase the study N. The results suggest this piece, which uses logical methodology but is low powered, is more appropriate as a preliminary report, i.e., a Brief Report.

Response: These two paragraphs are related. the editor is giving us two solutions to the problems she perceives with the analysis, which all essentially related to the way we worked around the small sample size. given the time it would take to collect more data, perhaps following the second option would be the easiest; that is, reducing the word count as suggested and resubmitting as a brief report, acknowledging the small sample size as a limitation. regarding the comments: in truth the alpha of .10 was not determined in advance; since the purpose of the analysis is to understand the predictors of the dv, and not to build the most explanatory model, i disagree that the adjusted r2 is low; finally, i disagree it is inappropriate to talk about nonsignificant predictors in the discussion as this suggests an over-focus on p-values, which we know to be determined by sample size, meaning that an interpretation of effect sizes apart from p-values remains valid.

Author Response in the paper:

We wish to be transparent that the value of alpha was not set higher before the analysis; rather, we allowed this threshold to reflect marginal significance on reflection of the small sample size. We have made this explicit in the procedure “However, because this value was not set a priori we considered this as evidence of ‘marginal significance’ relative to the more traditional alpha of .05.”. This should also be evident in the way we discussed the finding in the discussion: “However, in our current study, intrinsic motivation did not emerge as a significant predictor of quality of life in breast cancer patients at the traditional alpha level of p < .05 (although can be considered marginally significant at the less conservative level of p < .10)”.

Regarding the value obtained for adjusted R2, we acknowledge that relative to the full amount of variance that can be explained 38% may appear low. However, within the social sciences, values of R2, which mathematically indicate the degree of relationship between one dependent variable and multiple predictors and are always larger than adjuster R2, reflect a moderate effect if they are greater than .25 (Ferguson, 2009). Thus, we argue that the value should not be considered low.

Please see that the small sample size has been included as a limitation in the discussion section: “Finally, we acknowledge that the sample size for the study was relatively small. Estimated p-values are dependent on sample size, meaning that when sample sizes are small, the likelihood of committing a Type II error are heightened. Although we attempted to counter this using a more alpha liberal threshold for determining significance, it is possible the analytical approach lacked statistical power to identify some true population effects. Thus, future studies with larger samples are required to replicate and confirm these findings.”

Such a move will require downsizing of the Discussion. The Discussion in any case should be broken into subsections, e.g., Relevant Psychological Variables, Limitations, Future Efforts. The last paragraph should be its own section — Conclusion.

Such a move will require downsizing of the Discussion. The Discussion in any case should be broken into subsections, e.g., Relevant Psychological Variables, Limitations, Future Efforts. The last paragraph should be its own section — Conclusion.

Response: We have introduced subsections in the discussion (Line 275 and 330), and we have added the Conclusions section (Line 361)

Specific Comments to Authors:

  1. 1, Para. 2, line 38:

and hormonal status -> physiological and hormonal status

Response: It has been modified.

  1. 2, Para. 1, line 43:

Add a sentence about protective factors in premenopausal women.

Response: Lines 48-50 contain information on protective factors in women. We have removed the phrase "post-menopausal" as it was an error, these protective factors refer to all women regardless of age.

  1. 3, Para. 3, lines 98-105:

State in these several sentences whether the women are patients, breast cancer patients, etc.

Response: Lines 102-112 specify the type of sample in each of the cited studies.

  1. 3, Para. 4:

Authors to consider whether the objectives paragraph should mention attitude towards physical activity, which the previous two paragraphs deal with.

Response: We did not introduce information on attitudes towards physical activity in the objectives, since it is not the objective of the study, and we did not investigate this variable either, what we analyzed are some psychological variables and the perception of the quality of life of the participants.

  1. 4, Para. 3, line 146:

How many points is the Likert scale?

Response: We have introduced the points of the Likert scale (Line 149).

  1. 5, Para. 3, line 184:

Supply a reference for the lFlS contention.

Response: We have added the reference IFIS, Ortega et al. [40]. on line 187.

  1. 5, Para. 5, lines 201-2:

Since it is doubtful the investigator can mention to every participant the importance of providing honest answers, suggest shortening this sentence to: “The principal investigator supervised the process.”

Response: Thanks for the suggestion, we have shortened the phrase (Line 205).

  1. 6, Para. 1, line 207:

List in parentheses the relevant table.

Response: We have entered information about the tables that correspond to each analysis (Lines 210, 214, 217 and 226).

  1. 6, Para. 2, line 214:
  • Add a few sentences on how the Pearson’s correlation helped guide choice of regression
  • What cut-off value was used to decide on the relevance of a variable?

  • List in parentheses the relevant

Response:

1 and 2) We acknowledge this could be elaborated. To be more specific, acknowledging sample size limitations we opted to excluded variables that were not significantly correlated with the DV or that could induce problems with multicollinearity. Basic needs and identified regulations were not significantly correlated with the DV, so excluded. Introjected regulations was strongly correlated with intrinsic motivation, and so based on theory we chose to retain intrinsic motivation. Moreover, the negatively related motivations all had significant and moderate correlations with self-esteem, so were also excluded. This resulted in a smaller model of positive predictors. Please see that we have summarized this in the text. No other specific cut-offs were used.

“Basic needs and identified regulation were not considered in the final model due a pattern of non-significant correlations. Moreover, integrated regulation, introjected regulation, external regulation and amotivation were excluded to avoid multicollinearity”.

3) We have entered information about the tables that correspond to each analysis (Lines 210, 214, 217 and 226).

  1. 6, Para. 2:

line 217: List in parentheses the relevant table.

Response: We have entered information about the tables that correspond to each analysis (Lines 210, 214, 217 and 226).

line 223: (1) Supply a reference supporting that use of a .10 alpha critical value is justifiable; (2) State, if it is the case, that a value of .10 was selected in advance of the analyses; and (3) see above general comments on the 2 options recommended for this study/paper.

Response: Please see the following reference as an example of when different thresholds can be considered when sample sizes are small. In fact, in its original conception, the NHST approach never specified that .05 was the absolute thresholds for all analyses; indeed, they should be custom to the analyses being performed. But better than setting the threshold and having an all-or-nothing approach, it is preferred to present exact p-values to allow an interpretation of the probability of achieving the test statistic or greater if the null hypothesis were true in the population. This, coupled with interpretation of effect sizes, provides a much more sensitive understanding of the outcome of a statistical test.

We wish to be transparent that the value of alpha was not set higher before the analysis; rather, we allowed this threshold to reflect marginal significance on reflection of the small sample size. We have made this explicit in the procedure “However, because this value was not set a priori we considered this as evidence of ‘marginal significance’ relative to the more traditional alpha of .05.”.

Lee, E.C.; Whitehead, A.L.; Jacques, R.M.; Julious, S.A. The statistical interpretation of pilot trials: should significance thresholds be reconsidered?. BMC Med Res Methodol, 2014, 14(1), 1-8. https://doi.org/10.1186/1471-2288-14-41

  1. 7, Table 3, line 247:

Given that the Discussion includes mention of the variable “integrated regulation,” consider extending Table 3. to the next three p-value relevant variables (in terms of magnitude) if “integrated regulation” is among them.

Response: The discussion does mention integrated regulation, but in relation to the correlational analysis. We agree it would be interesting to include all the other variables into the regression analysis, but the limitation of the sample size, and issues with multicollinearity preclude it. However, thank you for the suggestion.

  1. 8, Para. 2, line 254:
  2. Discussion -> 4. Discussion

 Response: The numbering of sections and subsections has been revised and modified.

  1. 8, Para. 2, line 255:

The Discussion is long. Suggest adding a subsection title, 4.1. Relevant Psychological Variables

Response: We have introduced subsections in the discussion (Line 275 and 330), and we have added the Conclusions section (Line 361)

  1. 8, Para. 5, line 277:

Demotivation and competence would seem to move in opposite. Can this language be clarified or rectified for the reader?

Response: We have redacted that information (Line 296-297)

  1. 8, Para. 6, line 286:

Add a sentence explaining how the study’s findings can suggest the limited confidence and understanding indicated.

  1. 9, Para. 5:

line 312: Add subtitle: 4.2. Limitations

Response: We have introduced subsections in the discussion (Line 275 and 330), and we have added the Conclusions section (Line 361)

line 322: Address how moderate sample size may have affected study results, i.e., model .38 R2 adjusted, need for .10 alpha critical value in the regression.

Response: Please see that we have added the following text to the discussion as a way to consider the limitations of the small sample. Finally, we acknowledge that the sample size for the study was relatively small. Estimated p-values are dependent on sample size, meaning that when sample sizes are small, the likelihood of committing a Type II error are heightened. Although we attempted to counter this using a more alpha liberal threshold for determining significance, it is possible the analytical approach lacked statistical power to identify some true population effects. Thus, future studies with larger samples are required to replicate and confirm these findings.”

  1. 9, Para. 6, line 285:

Add subtitle: 4.3. Future Efforts

Response: We have introduced subsections in the discussion (Line 275 and 330), and we have added the Conclusions section (Line 361)

  1. 10, Para. 1, line 335:

Add a second paragraph to this section describing how sample size could be increased.

 Response: This study is part of a larger project and in the next phase the second sample will be expanded. We must remember that these are participants with very specific characteristics, they are women with breast cancer from the province of Cáceres who are receiving chemotherapy, radiotherapy and/or hormonal treatment at the Hospital San Pedro de Álcantara in Cáceres. Hence the complexity of recruiting the sample.

  1. 10, Para. 2, line 336:

Add section title: 5. Conclusion

Response: We have introduced subsections in the discussion (Line 275 and 330), and we have added the Conclusions section (Line 361)

Grammatical Changes:

  1. 2, Para. 1, line 45:

post-menopausal -> postmenopausal

Response: We have removed that term.

  1. 2, Para. 2:

line 59: as well as body mass index -> as well as with respect to body mass index

Response: It has been modified (Line 63)

lines 62-3: in treatment side effects [19,20]. Therefore, there is evidence -> in treatment side-effects [19,20]. In summary, there is evidence

Response: It has been modified (Line 67)

  1. 2, Para. 3, line 76:

as well as experiencing -> as well as to experience

Response: It has been modified (Line 80)

  1. 3, Para. 2, line 93:

self-determined motivation, it is associated -> self-determined motivation. This condition is associated -or- self-determined motivation. This status is associated

Response: It has been modified (Line 97)

  1. 3, Para. 3:

line 96: There is a limited number of studies that have applied -> A limited number of studies have applied

Response: It has been modified (Line 100)

line 98: For instance, Milne et al. -> Milne et at.

Response: It has been modified (Line 102)

  1. 3, Para. 6, line 121:

M -> mean (M)

Response: Within the context we consider that it is not necessary to specify that the M is mean, like the standard deviation (SD)

  1. 4, Para. 2, line 132:
  • enrollment -> and 4) enrollment

Response: It has been modified (Line 135)

Para. 3, Para. 3:

line 142: , and emotional well-being -> , emotional well-being

functional well-being -> and functional well-being

Response: It has been modified (Line 145)

  1. 4, Para. 4, line 153:

18 items, organized -> 18 items organized

Response: It has been modified (Line 157)

  1. 6, Para. 2, line 218:

VIF values -> <speII out acronym> (VIF) values

Response: It has been modified (Line 225)

  1. 10, Para. 2, line 336:

In conclusion, -> In the regression (no comma after “regression”)

Response: It has been modified (Line 362)

Round 2

Reviewer 1 Report

My comments have been well addressed. Thank you very much for the revisions.

Author Response

My comments have been well addressed. Thank you very much for the revisions.

RESPONSE: we would like to thank all the comments and suggestions.

Reviewer 2 Report

See enclosed comments.

See enclosed comments.

Author Response

General Comments to Authors:

The reviewer thanks the authors for accepting the bulk of the suggestions offered. Where a suggested change was not incorporated, the authors provided an adequate explanation. The reviewer appreciates the authors’ point of view on the adjusted R2 and P-value threshold and that they have provided two references in support of their positions, while recognizing that more conservative perspectives also exist on their choice of threshold values. The final predictor variables the authors have chosen make sense with respect to the hypothesis they are exploring, and the statistical approach is appropriate, thus the paper in proper, reduced form should be published.

As the authors note, their article is an exploratory piece to search out variables. Future careful enlargement of the study sample will no doubt firm up the authors’ findings. The chances of a false positive in the final variables accepted will decrease, and the number of statistically significant variables will most likely increase. At its current stage the piece needs to be tailored down to a Brief Report, paper size 2,500 words or less. The authors need to pursue this option.

Response: We would like to thank all the comments and suggestions. This paper is part of a larger project, and we consider it interesting to publish these relevant data for the scientific community as a scientific paper, given the special characteristics of the sample.

Specific Comments to Authors:

  1. 2, Para. 1, line 44:

and psychological and hormonal status ->   and physiological hormonal status

The simple term “hormonal status” cannot be used because 2 lines earlier the authors clarify they are describing non-modifiable risk factors. Contraceptives modify hormonal status, thus a qualifying term like “physiological” or “natural” needs to be placed in front of “hormonal status.”

Response: It has been modified (Line 44).

  1. 6, Para. 1, line 214:

2023) (Table 1).. ->   2023) (Table 1).

Response: It has been modified (Line 214).

  1. 6, Para. 3, line 231:

considered this as evidence ->   considered such findings

Response: It has been modified (Line 231).

  1. 9, Para. 1, lines 296-7:

demotivation toward the practice of physical activity and the satisfaction of the BPN competence were prevalent. -> lack of motivation (demotivation) for undertaking physical activity and towards satisfying certain basic psychological needs (BPNs) such as feelings of competence and autonomy were commonly observed.

Sentence is both ambiguous and hard to interpret for the reader. I suggest the above clarification or similar.

Response: It has been modified (Line 296).